# Superhydrophobic Methylated Silica Sol for Effective Oil–Water Separation

**DOI:** 10.3390/ma13040842

**Published:** 2020-02-13

**Authors:** Jiao Li, Hao Ding, Heqiang Zhang, Chunlin Guo, Xiaoyan Hong, Luyi Sun, Fuchuan Ding

**Affiliations:** 1College of Chemistry and Materials Science & Fujian Key Laboratory of Polymer Science, Fujian Normal University, Fuzhou 350007, China; lijiao277@163.com (J.L.); huangfuheqiang@outlook.com (H.Z.); guochunlin2020@163.com (C.G.); hongxiaoyan123@163.com (X.H.); 2Polymer Program, Institute of Materials Science, University of Connecticut, Storrs, CT 06269, USA; hao.2.ding@uconn.edu; 3Department of Chemical and Biomolecular Engineering, University of Connecticut, Storrs, CT 06269, USA

**Keywords:** methylated silica, superhydrophobicity, oil–water separation

## Abstract

Superhydrophobic methylated silica with a core–shell structure was successfully fabricated by a sol-gel process. First, a pristine silica gel with an average particle size of ca. 110 nm was prepared, using tetraethylorthosilicate (TEOS) as a precursor, ethanol as a solvent, and NH_4_OH as a catalyst. Then, the superhydrophobic methylated silica sol was prepared by introducing methyltrimethoxysilane (MTMS), to graft the surface of the pristine silica gel with methyl groups. The structure and morphology of the methylated silica sol were characterized by Fourier transform infrared (FTIR) spectroscopy, field emission scanning electron microscopy (FE-SEM), and transmission electron microscope (TEM). The characterization results showed that methyl groups were successfully grafted onto the surface of the pristine silica, and the diameter of the methylated silica was increased by 5–10 nm. Various superhydrophobic surfaces on glass, polyethylene terephthalate (PET) fabric, cotton, open-cell polyurethane (PU) foam, and polypropylene (PP) filter cloth were successfully constructed by coating the above substrates with the methylated silica sol and reached with a maximum static water contact angle and slide angle of 161° and 3°, respectively. In particular, the superhydrophobic PP filter cloth exhibited promising application in oil–water separation. The separation efficiency of different oil–water mixtures was higher than 96% and could be repeated at least 15 times.

## 1. Introduction

Superhydrophobic surfaces with a water contact angle (CA) greater than 150° and a sliding angle smaller than 10° have attracted huge interest for both fundamental research and practical applications [1], such as anti-sticking, anti-contamination, self-cleaning, and oil–water separation [2,3,4,5,6,7,8]. 

The two basic principles of constructing superhydrophobic surfaces are suitable surface roughness and surface energy, which can be typically tuned by surface treatment [9,10]. Based on such principles, many methods have been developed to prepare artificial superhydrophobic surfaces, including vapor deposition [11,12], etching [13,14], layer-by-layer assembly [15], self-assembly [16,17], electrospinning [18,19], phase separation [20], and sol-gel method [21,22,23]. 

Colloidal silica spheres have been widely used in many important industrial products, including catalysts, pigments, personal care products, and pharmaceuticals [24,25]. Spherical silica nanoparticles can be synthesized by a sol-gel method from alcohol solutions of silicon alkoxides in the presence of ammonium hydroxide as a catalyst [26]. However, pristine silica is hydrophilic and uptakes water due to the presence of Si–OH groups on the surface. Substituting the surface Si–OH groups by Si–O–R (where R = alkyl or aryl) groups would change the pristine silica from hydrophilic to hydrophobic [27,28]. Nevertheless, in most cases, expensive and toxic fluorinated chemicals are used to modify silica [29,30,31,32,33,34,35,36,37]. Therefore, the development of non-fluorinated modifying agents is very important for the fabrication of environmentally friendly coatings.

Oil contamination of water not only induces severe water pollution, but also threatens the health of human beings and all other living species in the ecological system. Oil–water separation is critical for the treatment of oily wastewater and oil-spill accidents. Traditional techniques such as oil skimming, centrifuge, coalesce, settling, filtration, and flotation technology are helpful in separating immiscible oil–water mixtures, but these methods are usually energy-consuming and not efficient [38]. Compared to the traditional separation techniques, materials with tunable superhydrophobicity and superoleophilicity for oil–water separation have attracted great interest thanks to their high efficiency and low operational cost [39,40,41,42].

Herein, we propose a new approach to prepare a superhydrophobic silica sol by sol-gel method, which can be used to generate a surface both superhydrophobic and superoleophilic, on soft and hard substrates, by simple dip-coating or the doctor-blading method. It is particularly interesting that the treated soft and hard materials exhibit outstanding water resistance and oil–water separation.

## 2. Experimental

### 2.1. Materials

Tetraethylorthosilicate (TEOS, 99%) and methyltrimethoxysilane (MTMS, 99%) were obtained from Aldrich. Ethanol (EtOH, 99.5%), aqueous ammonia (NH_3_·H_2_O, 25 wt.% in water), and dichloromethane were purchased from Sinopharm Group Chemical Reagent Co., Ltd. (Shanghai, China). PET dust-free fabric (3009) and polypropylene filter cloth (mesh number 400) were bought from Shenzhen Weike Anti-Static Technology Company (Shenzhen, China). Open-cell polyurethane (PU) foam with a density of 25 kg/m^3^ was obtained from Huntsman Corporation. PET films were purchased from Wantai Electronic Materials Co., Ltd. (Dongguan, China), and medical cotton cloth (Guangdong Fufeng Sanyuan Industrial Co., Ltd.) was purchased from a local drugstore. All chemicals were used directly, without further purification.

### 2.2. Sample Preparation

Pristine silica sol was synthesized, using a mixture of TEOS, EtOH, and H_2_O at a molar ratio of 0.55:1.3:0.02 by a sol-gel process via the Stöber method, with NH_3_·H_2_O as the catalyst [43]. In brief, 10.0 g of deionized water was mixed with 60.0 g of ethanol, and then 6.0 g of TEOS was diluted into the abovementioned solution, and the resultant solution was stirred for 10 min. Then 2.0 g of NH_3_·H_2_O (25 wt.% in water) was added as a catalyst. The solution was mechanically stirred at 60 °C for 2 h, to prepare pristine silica sol.

After the pristine silica was prepared, 2.0 g of deionized water and 1.2 g of methyltrimethoxysilane (MTMS) were added into 20.0 mL of the prepared pristine silica sol, and the mixture was stirred vigorously at 60 °C for 30 min. Under stirring, 0.2 g of NH_3_·H_2_O (25 wt.% in water) was added as a catalyst. After 12 h of reaction, a methylated silica sol was prepared, which was further diluted to 0.5 wt.% for coating.

The pristine and methylated silicas were coated on PET films by doctor blading with a thickness of 1.5–2.0 µm. Glass slides were coated by both the pristine and methylated silicas by dipping the slides into the prepared sols for 30 s, and then they were withdrawn at a speed of 60 mm/min. The methylated silica was impregnated into PET fabric, medical cotton, PU foam, and PP filter cloth by a direct soaking process (5 min of soaking time). All coatings were dried at 80 °C for 10 min, to evaporate the solvent. The weight uptakes of the coatings for the PET fabric, medical cotton, PU foam, and PP filter cloth are ca. 2.5%, 3.5%, 2.0%, and 1.0%, respectively.

### 2.3. Characterization

The ultraviolet-visible (UV-Vis) spectra of the PET film, with and without coating, were recorded using a UV-Vis spectrophotometer (Lambada 900, PerkinElmer, Inc., Massachusetts, United States. The surface functional groups of the superhydrophobic silica sol were analyzed by a Fourier transform infrared (FTIR, Nicolet 5700, Thermo Electron Scientific Instruments Corp., Massachusetts, United States) spectrophotometer. The thermal degradation of the silica was characterized by a thermogravimetric analyzer (TGA, model Q600, TA Instruments, Delaware, Unite State), under a nitrogen atmosphere, at a heating rate of 10 °C/min.

The contact angle (CA) and slide angle (SA) of the samples were measured, using a contact angle analyzer (DSA-25, German Kruss Scientific Instrument Co., Ltd., Hamburg, German), at 25 °C, with 6 µL of deionized water or hexadecane droplets on the sample surface. The CA was measured 10 times at different locations, and an average of the five data within ±2° were recorded for each sample. 

The particle shape and size of the silicas were characterized by a field emission scanning electron microscope (SEM, 7800F, JEOL Ltd., Tokyo, Japan). The morphology of single silica particles was observed by transmission electron microscopy (TEM, JEM-2100F, JEOL Ltd., Tokyo, Japan). The silica sols of 0.1 wt.% were dropped on copper grids coated with carbon and dried at room temperature for TEM imaging.

## 3. Results and Discussion

### 3.1. Reaction Mechanism of Superhydrophobic Silica Sol

As Figure 1 shows, methylated silica sol was prepared by a facile surface treatment, according to the Stöber method [43]. In the first step, pristine silica gel was prepared by a sol-gel method, using tetraethylorthosilicate (TEOS) as the precursor, ethanol as the solvent, and NH_4_OH as the catalyst. Under base catalysis, Si(OH)_4_ was produced by hydrolyzing the ethoxy groups of TEOS [44,45]. Pristine silica sol was then prepared through de-hydration and de-alcohol condensation between TEOS/partially hydrolyzed TEOS and pristine silica. Then, methylated silica sol was prepared by reacting with MTMS to graft methyl groups on the silica surface. MTMS was selected as the modifier here because it has one methyl and three methoxy functional groups. When MTMS was added to the pristine silica sol, de-hydration and de-alcohol took place under the catalysis of ammonia. As a result, most hydrophilic hydroxyl groups on the silica surface were converted to hydrophobic methyl groups [46].

The chemical composition of the pristine and methylated silica was confirmed by FTIR, as shown in Figure 2a. The pristine silica and methylated silica sol show two strong absorption bands at 1095 and 452 cm^−1^, which can be assigned to the stretching and bending vibration of Si–O–Si bonds, respectively [47]. The absorption bands of 3340 and 954 cm^−1^ for Si–OH groups and 1640 cm^−1^ for the adsorbed water decreased significantly after MTMS modification. In the spectrum of the methylated silica sol, the peaks at 2975 and 1274 cm^−1^ can be attributed to the C–H in –CH_3_ groups and Si–C groups, respectively [46]. In brief, the FTIR characterization supports that MTMS molecules were successfully grafted onto the surface of silica, producing silica–CH_3_ surface functional groups.

As shown in Figure 2b, the TGA thermograms demonstrated a clearly different thermal degradation between the pristine and methylated silica. For the pristine silica, the first weight loss step from 30 to 150 °C and the second weight loss step from 300 to 600 °C were attributed to the evaporation of the adsorbed water and the dehydration condensation between –OH groups, respectively. Different from the pristine silica, due to the superhydrophobic nature, there is virtually no water loss in the methylated silica before 100 °C. The weight loss from 100 to 600 °C results from the combination of the dehydration condensation between the –OH groups and the degradation of the –CH_3_ groups [44]. From the weight-loss steps of the methylated silica, the graft ratio of –CH_3_ groups is estimated to be ca. 8 wt.%.

### 3.2. Morphology of the Pristine and Methylated Silica Coatings

For the formation of a superhydrophobic coating, usually, both the modification of surface chemistry and surface roughness enhancement are critical [48], with surface morphology typically as the more dominating factor. The resultant superhydrophobic silica sol was used to fabricate coatings on various substrates. To observe the morphology of the silica particles, the synthesized sol was diluted to 0.1 wt.%, to avoid particle aggregation. Figure 3a shows the pristine silica particles of ca. 100 nm in diameter, with a narrow size distribution and virtually no aggregation. After MTMS treatment, the formed methylated silica was covered with a thin layer of 5~10 nm (inset of Figure 3b). The surface of the methylated silica is rougher than the pristine silica. Figure 3c shows the surface morphology of the methylated silica coating on a PET film, whose surface is rugged, with each domain composed of many small particles. The image indicates that a high level of porosity on the coating surface was formed. Overall, the characterization results suggest that the methylated silica coating has a hierarchical micro/nano structure, similar to the structure of the lotus leaf [49], and these hierarchical pinecone-like structures generate numerous grooves, in which air can be trapped, leading to a more hydrophobic surface [44]. At the same time, the surface was covered with a layer of low polarity –CH_3_ groups. The formation of a rough surface with low surface energy is expected to endow the coating with both superhydrophobic and superoleophilic properties [50].

Figure 4 shows the transparency of the PET films coated with pristine or methylated silica sol. In general, hydrophobicity and transparency are conflicted from the viewpoint of surface roughness, because a rough surface typically induces light scattering, and thus decreases transparency. Figure 4 shows that the PET film coated with the pristine silica exhibits a high transparency (>90% at 400~800 nm), virtually the same as the non-coated PET. The PET film coated with the methylated silica sol exhibits a slightly lower transparency because of the rough surface, as shown in Figure 3c. However, the reduction in transparency is very marginal, which is probably because of the very low surface roughness of the methylated silica coating in the micro-/nanometer range. This is very desirable for applications requiring both high transparency and superhydrophobicity.

### 3.3. Wetting Behavior and Corresponding Application Demonstrations

To evaluate the wetting behavior of the coatings, the images of water droplets on the coatings on glass slides are shown in Figure 5. After MTMS modification, polar (–OH) groups were replaced by nonpolar and thermally stable –CH_3_ groups, forming a rough low-surface-energy superhydrophobic and superoleophilic surface [47]. The static water CA was changed from 13.5° (pristine silica coating, Figure 5a) to 161° (methylated silica coating, Figure 5b). Figure 5c shows a digital picture of water drops on a glass slide coated with methylated silica. The virtually spherical drops proved the superhydrophobicity of the methylated silica coating. The hexadecane CA of the methylated silica coating was tested to be only 4.5°, which shows that the methylated silica coating is superoleophilic (Figure 5d). Figure 6 and Appendix A show that a drop of water can roll and bounce on the surface of the methylated silica coating on a glass slide, and the water slide angle (WSA) is only 3°. The photos captured from 0 to 1.50 s in Figure 6 show the bouncing of a drop of water on the surface of the coating, which distinctly demonstrates the water repellence of the coating.

Superhydrophobic and superoleophilic surfaces were also achieved on PET fabric, medical cotton, PU form, and PP filter cloth by impregnating the methylated silica into their lacuna, and they were subsequently dried at 80 °C for 10 min, to remove the solvent. The SEM images of the methylated silica coating on the PP filter cloth are shown in Figure 7. As shown in Figure 7a,b, the surface of the PP filter cloth was coated with a layer of methylated silica. Since methyl groups of the methylated silica improve the interaction between the methylated silica and the PP surface, it enhances the adhesion between them and thus the durability of the treated filter cloth [51]. As Figure 7c shows, the surface of the PP filter cloth is not flat, since the coating was fabricated by a facile dipping process. Similar morphology was observed on the surface of PET fabrics and PU foams. The overall morphology of the coatings is of multilevel roughness, which is beneficial for achieving superhydrophobicity [48,52].

In order to test the performance of the methylated silica coating on various surfaces of a polar polymer, biomaterial, elastomer, and nonpolar polymer, the methylated silica was coated on the surfaces of a PET fabric, medical cotton, PU foam, and PP filter cloth by a direct soaking and drying process. Water (dyed to blue color by methyl blue) and dichloromethane (dyed to red color by tony red) are used as polar and nonpolar solvents (oil), respectively. As Figure 8a and Appendix A show, when placed on water, the methylated-silica-treated PET fabric and medical cotton could not be wetted at all and could float on the water’s surface. In strong contrast, the untreated samples were quickly wetted, and they precipitated into the water, after absorbing water. Due to their superhydrophobicity, when the methylated silica treated samples were inserted into water, a negative meniscus was formed on the solid–liquid–vapor interfaces that would prevent water from entering the substrate [50].

Figure 8b and Appendix A demonstrate the performance of the superhydrophobicity and superoleophilicity of the methylated silica coating. The treated PET fabric repelled water and absorbed dichloromethane immediately, while the untreated PET fabric absorbed both water and dichloromethane quickly. 

As Figure 8c and Appendix A show, a methylated-silica-coated PU foam cube could absorb virtually all dichloromethane from a dichloromethane–water mixture (vol/vol = 1:1), which can be potentially applied to recycle spilled oil from water. Similarly, the application of the superhydrophobic and superoleophilic methylated silica coating on a PP filter cloth for oil–water separation is presented in Figure 8d and Appendix A. When a mixture of dichloromethane–water (vol/vol = 1:1) was poured into the separator, virtually all dichloromethane (20 mL) quickly permeated through the treated PP filter cloth by gravity, and virtually all water (20 mL) was retained. The efficiency of oil–water separation for a series of different oil–water mixtures (vol/vol = 1:1), including vegetable oil, hexane, gasoline, petroleum ether, and dodecane, were studied. As shown in Figure 9a, the separation efficiency was higher than 96.7% for all of the abovementioned oil–water mixtures, suggesting that the coating can be used for the separation of a broad range of oils from water. After oil–water separation, the treated PP filter cloth was recycled, to test the reusability by separating the mixture of dichloromethane–water (vol/vol = 1:1). As Figure 9b shows, the oil-separation efficiency remained higher than 96.3% after 15 cycles of operation. The testing result indicates that the methylated silica coating has superhydrophobic and superoleophilic performance, with a high separation efficiency and excellent durability for oil–water separation. All of these result from the hierarchical structure and nonpolar surface of the methylated silica.

## 4. Conclusions

In summary, a convenient, economical, environmentally friendly, and scalable method to prepare multifunctional superhydrophobic and superoleophilic silica sol was reported. Superhydrophobic and superoleophilic surfaces can be directly constructed on hard and soft substrates by facile coating methods. The treated substrates exhibited excellent performance of water repellence, oil absorption, and oil–water separation. These properties can be practically applied in related fields.

## Figures and Tables

**Figure 1 materials-13-00842-f001:**
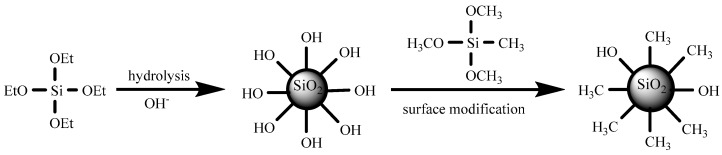
Schematic of the preparation of the methylated silica sol.

**Figure 2 materials-13-00842-f002:**
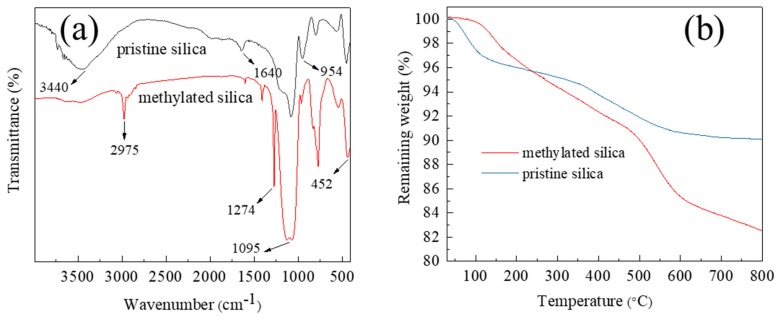
(**a**) FTIR spectra of the pristine and methylated silica. (**b**) Thermogravimetric analysis (TGA) of the pristine and methylated silica.

**Figure 3 materials-13-00842-f003:**
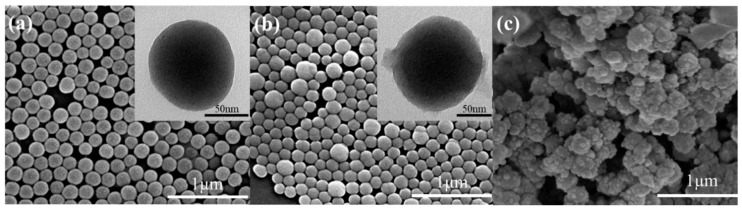
SEM and TEM images of the pristine and methylated silicas: (**a**) pristine silica, (**b**) methylated silica, and (**c**) methylated silica coating.

**Figure 4 materials-13-00842-f004:**
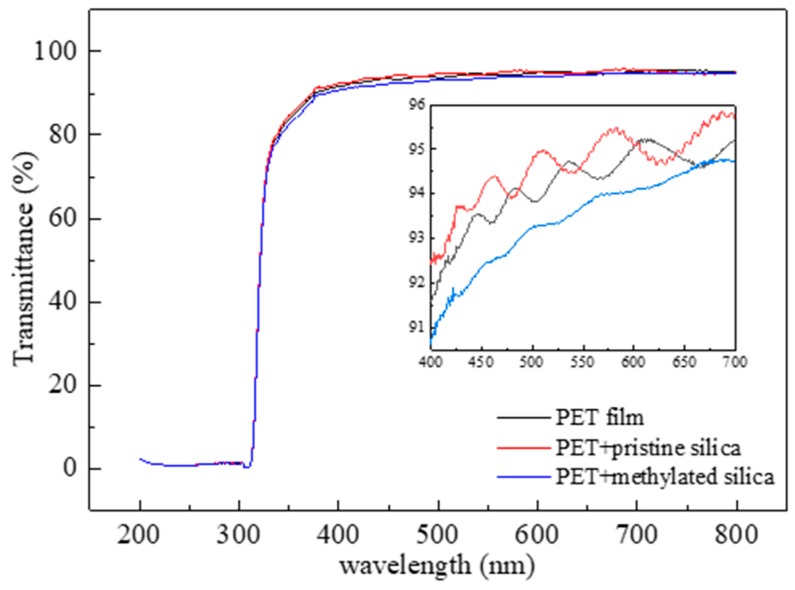
Transmittance of the coated PET films with pristine and methylated silica sol.

**Figure 5 materials-13-00842-f005:**
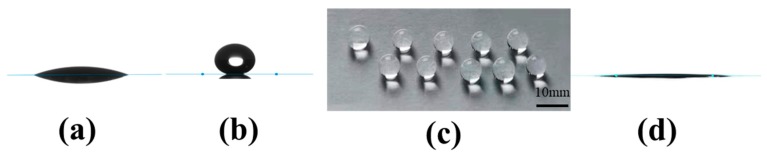
Water and hexadecane CAs of the pristine silica and methylated superhydrophobic silica coatings. (**a**) Water CA on the pristine silica coating, (**b**) water CA on the methylated silica coating, (**c**) digital photo of water droplets on the surface of a methylated-silica-treated glass slide, and (**d**) hexadecane CA on the methylated silica coating.

**Figure 6 materials-13-00842-f006:**
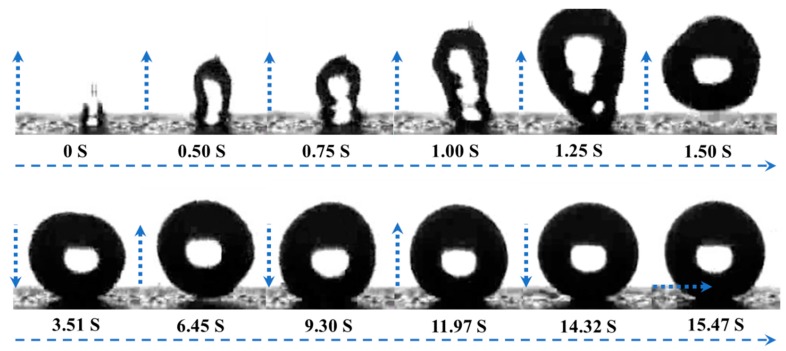
Bouncing of a water droplet on the surface of a methylated silica coating.

**Figure 7 materials-13-00842-f007:**
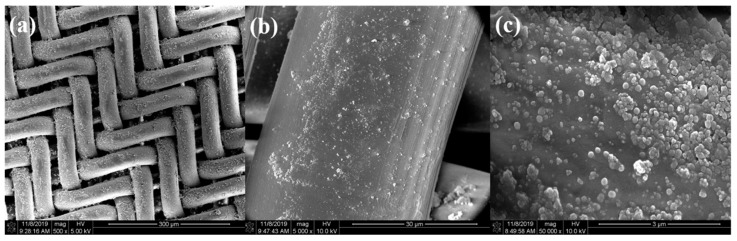
SEM images of the methylated silica coating on a PP filter cloth. (**a**) PP filter cloth coated with methylated silica. (**b**) A single thread of PP filter coated with methylated silica. (**c**) The methylated silica coating on surface of the PP filter.

**Figure 8 materials-13-00842-f008:**
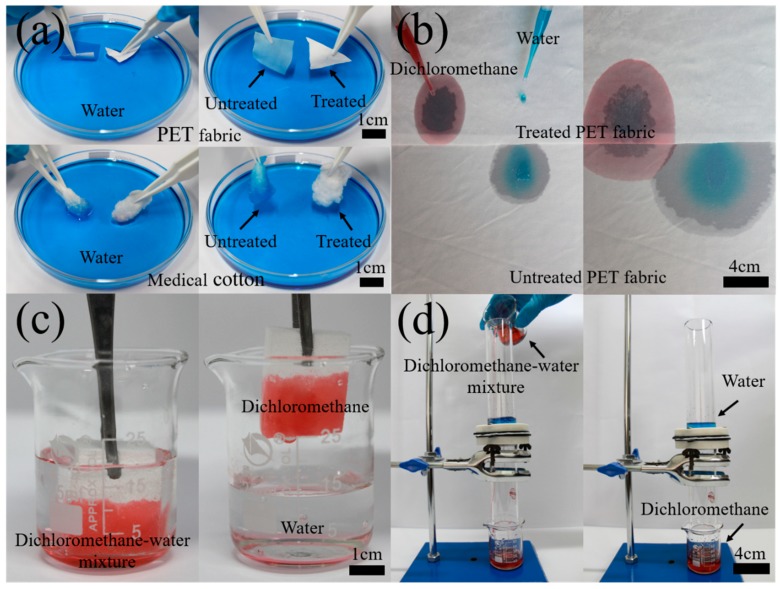
Applications of the superhydrophobic coating. (**a**) Comparison of water-repellent ability of the treated (left) and untreated (right) PET fabric and medical cotton. (**b**) Comparison of the super-wettability of treated (top) and untreated PET fabric (bottom) for dichloromethane (dyed red) and water (dyed blue). (**c**) Treated PU foam absorbing dichloromethane (dyed red) only in dichloromethane–water mixture. (**d**) Gravity-driven separation of dichloromethane (dyed red)–water (dyed blue) mixture by the treated PP filter cloth.

**Figure 9 materials-13-00842-f009:**
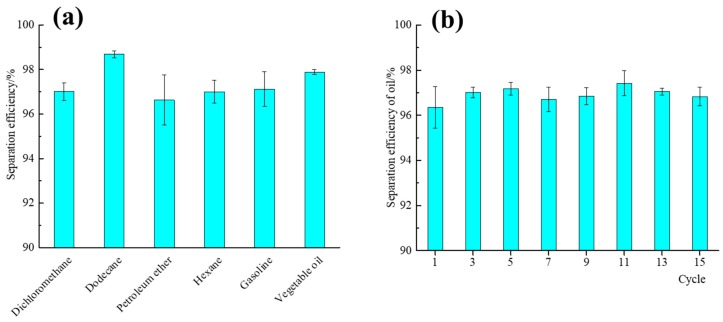
(**a**) Separation efficiency of the treated PP filter cloth for various oil–water mixtures. (**b**) Separation efficiency of the treated PP filter cloth for dichloromethane–water mixture after 15 cycles.

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
