# Peer review of "Superhydrophobic Methylated Silica Sol for Effective Oil–Water Separation"

_materials, 2020, doi:10.3390/ma13040842_

Round 1

Reviewer 1 Report

The manuscript deals with the synthesis of superhydrophobic nanoparticles by using the Stober method. A suitable coupling agent was added to the standard method in order to confer the superhydrophobic behavior to the produced particles. The described method, being free of Fluorine, appears very interesting and promising. The manuscript is well structured and the read is fluidic and clear. However, in my opinion, some minor revisions should be performed.

Line 104: Figure 1: the reaction scheme does not appear correctly; it is suggested to substitute it. Line 133: about the FT-IR, the authors are invited to better explain the method employed to estimate the graft ratio of –CH3. Do the authors have evidence of the coating’s adhesion on the substrate? It could be interesting to report on it. Line 167: Figure 4: please, insert an enlargement of the curved between 400 and 700 nm, in order to highlight the little differences in transmittance. Line 188: please, add some other comments about the Fig. 7 results, maybe also adding an EDS to clarify the elements nature. Line 199: please do you mind to give evidence (by inserting a photo) of the formation of the negative meniscus? Lines 195-219 and Figure 8: in the test, please better explain the experiments, the employed liquids and the nature of the treated and untreated samples (e.g., PET or cotton). Consequently, in Figure 8, add more labels for clarity. Line 214: please explain the method employed to assess the separation efficiency (to be add to the Experimental- Characterization paragraph) Please discuss better and longer introduction, results and conclusions.

Reviewer 2 Report

This paper reports on new silica sol exhibiting superhydrophobic property

and application for O/W separation.

Including video data, results are presented clearly to support main points 

of significance and description (explanation) of this study. Hence, it should be accepted almost as it is.

At the proof stage, please improve Figure 1, which was incorrectly shown 

at least on my PC.

That's all.

Reviewer 3 Report

The manuscript includes the preparation of modified silica particle to use the surface modification. The manuscript is well written, but some points should be improved as shown follows,

1) In introduction, please mention the engineering motivation of oil/ water separation, that should be compared with other techniques.

2) please change Figure 1. it is hard to see.

3) Figure 6 should be deleted because no discussion here ( about the deformation and elasticity)

4) how to determine the concentration of oil. Was it gas chromatography? if so please mention to experimental

Round 2

Reviewer 1 Report

The manuscript is now able to be published.